# Growth and Expression of Virulence Genes of *Listeria monocytogenes* during the Processing of Dry-Cured Fermented “Salchichón” Manufactured with a Selected *Lactilactobacillus sakei*

**DOI:** 10.3390/biology10121258

**Published:** 2021-12-02

**Authors:** Irene Martín, Alberto Alía, Alicia Rodríguez, Francisco Gómez, Juan J. Córdoba

**Affiliations:** Food Hygiene and Safety, Meat and Meat Products Research Institute, Faculty of Veterinary Science, University of Extremadura, Avda. de las Ciencias, s/n, 10003 Cáceres, Spain; iremartint@unex.es (I.M.); albertoam@unex.es (A.A.); fgmzpo@unex.es (F.G.); jcordoba@unex.es (J.J.C.)

**Keywords:** *Listeria monocytogenes*, dry-cured fermented sausage, *Lactilactobacillus sakei*, virulence gene expression

## Abstract

**Simple Summary:**

During the ripening process of the dry-cured fermented sausage “salchichón”, *Listeria monocytogenes* could fail to be eliminated. In addition, the food safety criterion for *L. monocytogenes* in the European Union sets up a maximum level of 100 units of this microorganism per gram in ready-to-eat products throughout their shelf-life. Thus, since *L. monocytogenes* could be present in this product, it is necessary to evaluate the impact of the dry-cured fermented processing in the potential virulence of this pathogen, even considering the possible effect of the usual microbiota (lactic-acid bacteria) of “salchichón”. In this work, the effect of the processing of “salchichón”, inoculated with a selected strain of *Lactilactobacillus sakei,* on the growth of *L. monocytogenes* and on the expression of its virulence genes, was evaluated. The processing of “salchichón” provoked a relevant reduction in *L. monocytogenes*, but this pathogen was not completely eliminated. However, a downregulation in the expression of the tested virulence genes was found, which could suppose a reduction in the pathogenic effect of this microorganism. These findings could be of great interest to consider the dry-cured ripening of “salchichón” as a safe process to control the pathogen *L. monocytogenes*.

**Abstract:**

The effect of the dry-cured fermented processing of “salchichón” inoculated with a selected strain of *Lactilactobacillus sakei* (205) on the growth and transcriptional response of three virulence genes (*plcA, hly,* and *iap*) of *Listeria monocytogenes* was evaluated. For this, three different batches of “salchichón” were analyzed: batch B (inoculated only with *L. sakei*), batch L (inoculated only with *L. monocytogenes*), and batch L + B (inoculated with both microorganisms). Sausages were ripened for 90 days according to a traditional industrial process. The processing of “salchichón” provoked a reduction in *L. monocytogenes* counts of around 2 log CFU/g. The downregulation of the expression of the three genes was found at the end of ripening when the water activity (a_w_) of “salchichón” was <0.85 a_w_. The combined effect on the reduction in *L. monocytogenes* counts together with the downregulation in the expression of the virulence genes throughout the “salchichón” processing could be of great interest to control the hazard caused by the presence of this pathogenic bacterium.

## 1. Introduction

*L. monocytogenes* is a Gram-positive, facultative intracellular bacterium responsible for human listeriosis, one of the most significant foodborne diseases in industrialized countries [1] that has well-known adverse health effects [2]. *L. monocytogenes* may contaminate food products at different steps of the manufacturing process, since this organism is able to survive on equipment and in production facilities [3,4]. It has been also reported that *L. monocytogenes* can pull through stressful environments, such as low temperature, high acidity, and salt contents [2,5,6]. This ability is a serious concern for the dry-cured fermented sausage industry. This is because dry-cured fermented meat products elaborated with meat and fat, mixed with salt, nitrate and/or nitrite, sugar, and spices like black pepper, which is stuffed into a casing and subjected to fermentation and drying processes, form a nutrient-rich but still restricted ecological niche that can favour the growth of *L. monocytogenes* [7].

The dry-cured fermented sausage “salchichón” is one of the most typical Spanish dry-cured meat ready-to-eat (RTE) products, characterized by a bacterial fermentation process followed by a ripening period [8]. Although the consumption of dry-cured fermented sausages is considered at low risk for foodborne listeriosis [9], the presence of this pathogenic bacterium has been reported in ripened sausages [10,11,12], and in some cases, it has also been involved in listeriosis outbreaks [13]. 

During the ripening of dry-cured fermented sausages such as “salchichón”, a reduction of *L. monocytogenes* has been reported, but its presence has not been completely eliminated [14]. In addition, the food safety criterion for *L. monocytogenes* sets up a maximum level of 100 CFU/g for this pathogen in RTE-food products throughout their shelf-life [15].

Since *L. monocytogenes* can be present in the dry-cured fermented sausage “salchichón”, it is necessary to evaluate the impact of the dry-cured fermented processing on the potential virulence of this pathogen. The intracellular parasitism of *L. monocytogenes* requires the coordinated expression of several genes that encode virulence factors, such as the *plcA*, *hly*, and *iap* genes [16]. The transcriptional response of *L. monocytogenes* under a variety of pH, water activity (a_w_), temperature conditions [17,18], or upon exposure to protective cultures [19] could result in changes in the *L. monocytogenes* viability and virulence [20,21,22,23,24,25]. 

To evaluate changes in the *L. monocytogenes* virulence during the processing of “salchichón”, the usual microbiota present in this product should be considered. This is mainly composed of lactic-acid bacteria (LAB) such as *Lactilactobacillus* spp. together with yeast and moulds [26]. LAB in conjunction with fermentation and ripening processes contributes to the organoleptic characteristics of the products as well as their stability [27,28,29]. *Lactilactobacillus sakei* is the most frequently isolated LAB in meat products, in particular on dry-cured fermented sausages [8,30,31,32,33,34,35] whose main functions in sausage fermentation are the acidification and inhibition of undesired autochthonous microbiota [36]. This bacterium can grow and survive under the conditions encountered during meat storage and processing [37]. In fact, *L. sakei* possesses the ability to use nutrients encountered in meat [30]. Furthermore, it has been proposed as a protective culture against *L. monocytogenes* in dry-cured fermented sausages [38,39,40]. 

Current knowledge regarding the effect of an antagonistic *L. sakei* strain against *L. monocytogenes* should be integrated with information concerning the molecular response of the pathogen to such treatments. Therefore, the objective of this work was to investigate the effect of the processing of the dry-cured fermented sausage “salchichón”, inoculated with a selected strain of *L. sakei,* on the growth of *L. monocytogenes* and the transcriptional response of some of its virulence genes (*plcA*, *hly*, and *iap*).

## 2. Materials and Methods

### 2.1. Bacterial and Culture Conditions

To evaluate the effect of the processing of “salchichón” on the growth and expression of the key virulence genes of *L. monocytogenes*, a strain of serotype 4b isolated from dry-cured meat products was used, since this serotype is the most frequently involved in clinical cases of listeriosis. In addition, this serotype has been reported as the highest transcriptomic response under hostile environments [41]. Thus, the strain *L. monocytogenes* S7-2 (serotype 4b), belonging to the National Institute of Agricultural and Food Research and Technology (INIA) collection (Madrid, Spain), has been used for the inoculation of “salchichón”. 

To evaluate the effect on LAB against *L. monocytogenes,* a selected strain of *L. sakei* 205 from the Food Hygiene and Safety Culture Collection at the University of Extremadura was used. This strain was isolated from traditional dry-cured fermented sausages and selected by its antagonistic activity against *L. monocytogenes* in a sterilized “salchichón”-based agar and in the dry-cured fermented sausages “salchichón” [14]. Thus, in the present work, the effect of the selected *L. sakei* strain on the expression of the virulence genes of *L. monocytogenes* in a real food system (dry-cured fermented “salchichón”) in the presence of the natural microbiological population of this product was evaluated.

To prepare the *L. monocytogenes* and LAB inocula, 100 µL of a stock culture (stored in Brain Heart Infusion (BHI) broth (Pronadisa, Madrid Spain) containing 20 % (*w*/*v*) glycerol at −80 °C) was sub-cultured twice onto BHI broth at 37 °C for 24 h and Man Rogosa Sharpe (MRS) broth (Fisher Bioreagents, Belgium) at 30 °C for 48 h, respectively (Figure 1). At the end of the incubation period, a suspension containing ≈ 8.0 log CFU/mL cells was obtained and an aliquot of this was diluted in 1% (*w*/*v*) peptone water to reach final concentrations of ≈7.0 and 6.0 log CFU/mL for *L. monocytogenes* and *L. sakei*, respectively. Next, after centrifuging cultures at 10,000× *g* for 5 min, the supernatants were discarded, and the sediments were washed and resuspended in PBS to be used for the inoculation of the “salchichón” mix before stuffing. To check the level of inoculation, serial dilutions were plated onto Chromagar^TM^
*Listeria* agar plates (Scharlab, Madrid, Spain) and incubated at 37 °C for 48 h for *L. monocytogenes*, while for LAB, dilutions were spread on MRS (Oxoid, Basingstoke, UK) agar plates and anaerobically incubated at 30 °C for 72 h. In addition, the real initial levels (log CFU/g) of *L. monocytogenes* and *L. sakei* 205 on the sausages were determined at day 0 of processing.

### 2.2. Elaboration of Dry-Cured Fermented “Salchichón”

The “salchichón” sausages were elaborated in a pilot plant located at the Faculty of Veterinary of the University of Extremadura (Spain) according to the industrial processing of this product [14], the mixture composition proceeding as follows: minced Iberian pork meat (90%) and Iberian pig fatback (7%), with an addition of NaCl (1.8%), cane sugar (0.4%), potassium nitrate (120 ppm), sodium nitrite (100 ppm), black pepper, and spices (Figure 1). This mixture was divided into 3 different batches, to which 150 mL of PBS with the corresponding microorganism’s inoculum, prepared as has been described in Section 2.1, were added. The three batches were: batch B (control inoculated only with *L. sakei* at a concentration of ≈6 log CFU/g); batch L (inoculated only with *L. monocytogenes* at a concentration of ≈7 log CFU/g); batch L + B (inoculated with *L. monocytogenes* at a concentration of ≈7 log CFU/g combined with *L. sakei* at a concentration of ≈6 log CFU/g). All ingredients and corresponding inoculum were then mixed using an automatic kneader that was cleaned and sanitized between batches. The mixes of the different batches were then stuffed into regenerated collagen casings of 40 mm in diameter (Viscofan, Navarra, Spain), reaching each sausage an approximate weight of 500 g.

The sausages were ripened for 90 days in controlled drying chambers of the pilot plant of the Faculty of Veterinary Science following the conditions used in typical traditional processing of “salchichón” [14]: 5 °C at 85% relative humidity (RH) for 3 days, then 7 °C and 80% RH for the 17 days, 9 °C and 75% RH for 10 days, and 12 °C and 70% RH to reach 90-day ripening (Figure 1).

For physicochemical, microbial, and gene expression analyses, five sausages of each batch were taken at the beginning of processing (day 0) and at 15, 30, 45, 60, and 90 days of ripening. All of them were aseptically removed from the casings in a laminar flow cabinet (Telstar, Spain) before analytical determinations. Thus, the experiment consisting of 3 different batches × 5 sampling times × 5 different analyzed sausages/each batch and sampling time was evaluated once according to the European Union Reference Laboratory Technical Guidance Document for conducting shelf-life studies on *L. monocytogenes* in RTE foods (such as “salchichón”), where no growth or the growth probability of this pathogen is ≤10% [42].

### 2.3. Microbiological Analysis

At each sampling time, 10 g of each of the 5 dry-cured sausages “salchichón” were aseptically taken from the center of the product and mixed with 90 mL of 1% (*w*/*v*) peptone water and homogenized in a Stomacher machine (Seward, model 400 Circulator, West Sussex, UK) at 300 rpm for 1 min. One mL of this homogenate was frozen at −80 °C and stored for further RNA extraction and gene expression analysis. Another one mL aliquot was used to make serial dilutions for the estimation of *L. monocytogenes* and LAB counts. For this, decimal serial dilutions were subsequently carried out in 1% (*w*/*v*) of peptone water, and 100 µL of the cell suspensions were then spread onto the surface of the MRS agar and CHROMagar^TM^
*Listeria* Chromogenic medium to determine LAB counts and *L. monocytogenes*, respectively. Their incubation was performed for 24–48 h at 30 °C for LAB counts and for 24–48 h at 37 °C for *L. monocytogenes* counts. To evaluate the implantation of *L. sakei* 205 in all batches and at the last sampling time (90 days), 50% of the characteristic LAB colonies were randomly isolated from MRS plates and characterized by a sequencing analysis of the 16S rRNA region and PFGE analysis of the DNA with the restriction *NotI* and *SgsI* enzymes (Thermo Fisher Scientific, Waltham, MA, USA) [14]. 

### 2.4. RNA Extraction and Gene Expression Assay

One mL of each of the samples stored at −80 °C for gene expression analysis purposes was thawed in refrigeration (4 °C). Later, RNA was extracted according to the instructions of the MasterPure^TM^ Complete DNA and RNA Purification Kit (Epicentre, Madison, WI, USA), after centrifuging 1 mL aliquot (10,000 rpm, 10 min at 4 °C) and removing the resultant supernatant. To remove genomic DNA contamination, samples were treated with the RNAse-Free Dnase I (Epicentre). RNA concentration and purity (A_260/280_) were then measured using the Nanodrop^TM^ (Thermo Fisher Scientific). Complementary DNA (cDNA) was synthesized using about 500 ng of the total extracted RNA according to the manufacturer’s instructions for the PrimeScript^TM^ RT Reagent Kit (Takara Bio Inc., Kusatsu, Shiga, Japan). 

Quantitative PCR (qPCR) based on TaqMan^®^ methodology was then used to amplify the virulence-related genes of *L. monocytogenes*, *plcA*, *hly*, and *iap* following the method described by Alía et al. [17]. In addition, a RT-qPCR based on SYBR^®^ Green methodology was also performed to amplify the constitutive 16S rRNA gene used to ensure that both RNA extraction and cDNA synthesis processes were properly carried out according to Alía et al. [43]. The ViiA^TM^ 7 system (Applied Biosystems, Waltham, MA, USA) was used for qPCR performance. The reactions were prepared in MicroAmp^®^ Fast Optical 96-Well Reaction plates (Applied Biosystems). Five replicates of RNA from control samples (only *L. monocytogenes*) and template-free negative controls (ultra-pure water instead of cDNA) were also included in the runs. Data analysis on the absolute expression of the target genes were determined using ViiA^TM^ 7 V.1.2.2 Software (Thermo Fisher Scientific). The quantification cycle (Cq), the intersection between each fluorescence curve and a threshold line, was automatically calculated by the instrument using default parameters. The absolute expression levels of the three target genes from the different batches were extrapolated from the standard curves built for each gene as described by Alía et al. [17] by using the Cq values obtained for the samples. The absolute gene expression of each sample was evaluated in quintuplicate.

### 2.5. Physicochemical Analysis

The determination of pH and a_w_ was carried out only in the batch inoculated with *L. sakei* 205 (batch B). The pH value was recorded after homogenizing 3 g of each sample with 27 mL of distilled water with a digital pH-meter Crison Basic 20 (Crison, Barcelona, Spain), while a_w_ was determined on sausage slices using a Novasina Lab Master Water activity meter model AW SPRINT-TH 300 (Novasina AG, Lachen, Switzerland).

### 2.6. Statistical Analysis

Statistical analyses were performed using the software IBM SPSS Statistic for Windows v.22.0 (IBM, New York, NY, USA). The different batches and days of ripening were used as independent variables. The counts (Log CFU/g), a_w_, pH values, and absolute expression were analyzed as dependent variables. Once the dependent and independent variables of the analysis were determined, a study of the normality of the different data populations was carried out using the Shapiro Wilk test. The analysis of the data was conducted using the Mann–Whitney test [44], and the statistical significance was set at *p* ≤ 0.05.

## 3. Results and Discussion

### 3.1. Evolution of Water Activity and pH during Ripening of “Salchichón”

The a_w_ decreased (*p* ≤ 0.05) from an initial value of 0.947 a_w_ in the raw product to values below 0.790 a_w_ at day 90 (Figure 2). The evolution of a_w_ was similar to that reported for dry-cured fermented sausages by previous studies [45,46].

The evolution of pH throughout the ripening process of “salchichón” is shown in Figure 2. A decrease in pH values is observed after 15 days of ripening from 5.82 to 5.48 (*p* ≤ 0.05), probably due to the growth of the inoculated *L. sakei* 205. However, at day 30 of ripening, an increase in the pH value was observed (close to 5.8), and it then remained constant until the end of maturation. The increase in pH at the end of the ripening time may be due to the accumulation of non-protein nitrogen and amino acid catabolism products [47,48].

### 3.2. Evolution of Lactic-Acid Bacteria and L. monocytogenes Counts throughout the Ripening Process of “Salchichón”

The evolution of LAB counts in *L. sakei* 205-inoculated batches (batches B and L + B) was very similar, since levels varied between 6.5 and 7.5 log CFU/g in both batches during all the ripening times (Figure 3). There were no significant differences (*p* ≤ 0.05) in LAB counts between these two batches. However, in the batch inoculated only with *L. monocytogenes* (batch L), LAB counts were always lower than 6.5 log CFU/g, and for most of the ripening times were significantly lower (*p* ≤ 0.05) than those found in *L. sakei* 205-inoculated batches. These results were expected because of the inoculation with the selected *L. sakei* strain of dry-cured fermented sausages “salchichón” composing batches B and L + B. In batches inoculated with *L. sakei* (B and L + B), most of the tested isolates (86%) were identified as *L. sakei* (100% identity) by a sequencing analysis of the 16S rRNA region. In the PFGE analysis, these isolates showed the same pattern of *L. sakei* 205. The remaining strains were *Lactilactobacillus plantarum* group (7%) and *Lactilactobacillus curvatus* (7%). In batch L, none of the isolates were identified as *L. sakei*. Thus, the inoculated *L. sakei* 205 was the predominant strain in *L. sakei*-inoculated bathes (B and B + L) and it was not detected in the batch inoculated only with *L. monocytogenes* (L). 

Levels of *L. monocytogenes* decreased (*p* ≤ 0.05) in both inoculated batches (L and L + B) throughout the ripening process of the dry-cured fermented sausage “salchichón” (Figure 3). No increase in *L. monocytogenes* growth during the processing was observed in batches inoculated with this pathogen alone (L) or together with *L. sakei* 205 (L + B) (Figure 3), even considering that during the first 15 days of ripening, there could be certain conditions such as temperature (7 °C), a_w_ (0.947–0.914), and pH (5.8–5.4) that may allow the growth of *L. monocytogenes*. Likely, the synergistic effect of a_w_, temperature, and pH reduction together with the effect of NaCl and nitrite added, and the presence of LAB inoculated or from contamination, prevent the growth of *L. monocytogenes* in the first days of ripening [38,49,50]. After 15 days of processing, the a_w_ values of “salchichón”, ranging from 0.914 to 0.779 a_w_, did not allow the growth of *L. monocytogenes* [51,52]. From day 45 until the end of ripening, levels of *L. monocytogenes* were significantly lower in the batch inoculated with *L. sakei* 205 (L + B) than in the batch only inoculated with *L. monocytogenes* (L). This additional reduction in *L. monocytogenes* can be associated with the presence of the strain *L. sakei* 205. Different strains of this species have shown anti-microbial effects when they have been used as protective cultures [30,38,40]. In this work, a reduction of about 2 log CFU/g was achieved during the processing of dry-cured fermented sausage “salchichón”. This means that this pathogenic bacterium could be detected after 90 days of ripening if the levels of contamination in raw materials or during processing are higher than 2 log CFU/g. Thus, it is of great importance to evaluate the effect of processing in the expression of virulence genes of *L. monocytogenes*.

### 3.3. Effect of Processing and Presence of L. sakei on the Absolute Transcription Levels of L. monocytogenes Virulence Genes in Dry-Cured Fermented Sausages

Findings on this work revealed a high and significant relationship between the transcriptomic response and counts of *L. monocytogenes* (*r*-values: 0.792, 0.821, and 0.820 for the *plcA*, *hly*, and *iap* genes, respectively). The two virulence genes *plcA* and *hly* showed higher expression values than those found for the *iap* gene in all the tested conditions (*p* ≤ 0.05; Table 1). In addition, at 0, 15, and 30 days of processing, no differences (*p* > 0.05) in the gene expression values were found for all the analysed batches (Table 1). However, at days 60 and 90 of ripening, a significant decrease in the transcription levels regarding days 0, 15, and 30 was detected for all the analysed genes. 

These results showed that the virulence gene expression values decreased at the end of the ripening period, likely due to the stress conditions created due to the processing and composition of “salchichón” mainly characterized by the decrease in a_w_ throughout the ripening until values below 0.85 a_w_. A decrease in the *plcA*, *hly,* and *iap* gene expression values has been reported in a dry-cured ham model system as a consequence of a_w_ decrease [19]. Furthermore, a downregulation in the expression of *plcA* and *hly* genes has been reported in different stress conditions on dry-cured meat products, likely due to the *prfA* gene repression, the major regulator of *L. monocytogenes* virulence [6,53]. Thus, at the end of “salchichón” ripening with a_w_ below 0.85 a_w_, the expression of stress-related genes is likely enhanced, and the transcription of other non-stress-related genes such as virulence genes is repressed to facilitate the *L. monocytogenes* survival in highly stressful environments, as has been previously reported [6,54]. This agrees with the results obtained in this study, since it has been observed that the expression of the three virulence genes tested decreased. These findings are very interesting, since they correspond to real food samples. Most published works that showed an up-regulation in virulence genes of *L. monocytogenes* were in vitro-conducted with laboratory-based media [6]. However, the food matrix, especially a meat-based one, may influence the virulence potential of *L. monocytogenes*, possibly through the downregulation of virulence genes [6,21,55]. In addition, Bowman et al. [53] reported the suppression of *prfA* and *sigB* genes because of the non-thermal treatment on *L. monocytogenes*; this may also explain the decrease in the expression levels of *plcA* and *hly* genes.

When the effect of the inoculated *L. sakei* 205 on the expression of virulence genes tested was analyzed, the expression values of the three genes remained constant compared to those obtained in the control batch (L), except at day 60 of maturation, when significantly higher expression values were detected for the *hly* gene in batch L + B (Table 1). In addition, no differences between these two batches were found at the end of the ripening time (Table 1). These results contrast with the demonstrated effect of some bacteriocinogenic LAB such as *Enterococcus faecium* on the inhibition of the expression of the virulence genes of *L. monocytogenes* [56]. In the present work, *L. sakei* 205, which provokes an additional decrease in *L. monocytogenes* counts to those produced by the action of the reduction in pH and a_w_ throughout the ripening of “salchichón”, does not show any appreciable effect on the virulence gene expression of *L. monocytogenes* throughout this processing. To be used as a protective culture, it is important that *L. sakei* 205 does not cause an increase in the virulence gene expression of this pathogenic bacterium, since in some cases, various protective cultures such as selected strains of *Debaryomyces hansenii* have been reported to enhance the expression of virulence genes on *L. monocytogenes* [19].

In the present work, it has been demonstrated that a reduction in *L. monocytogenes* counts throughout the ripening process and a downregulation in the expression of virulence genes of the pathogenic bacterium cells could survive dry-cured fermented processing, but this effect was not increased by the addition of the strain *L. sakei* 205. The last effect is of great importance, since reports the relevance of the virulence gene transcription on the invasive and survival ability of *L. monocytogenes* in human models or cell lines [20,21,22,23,24]. Although further studies should be carried out to evaluate the effect of *L. monocytogenes* surviving “salchichón” ripening in culture cells, the processing of this product to reach a_w_ values below 0.85 a_w_ seems to be effective to control the microbial hazard caused by the presence of *L. monocytogenes*. 

## 4. Conclusions

This study describes for the first time the combined effect of the processing of “salchichón” manufactured with a selected protective culture of *L. sakei* on minimizing the growth and expression of three virulence genes (*plcA*, *hly*, and *iap*) of *L. monocytogenes*. The processing of “salchichón” provokes a reduction of 2 log CFU/g of *L. monocytogenes*. *L. sakei* 205 provokes an additional decrease in *L. monocytogenes* counts to those produced by the action of the reduction in pH and a_w_ throughout the ripening of “salchichón”. In addition, when the a_w_ of the product is lower than 0.85 a_w_, a downregulation in the expression of the above virulence genes was found. However, *L. sakei* 205 does not show any appreciable effect on the virulence gene expression of *L. monocytogenes* throughout “salchichón” processing. The combined effects of the reduction in *L. monocytogenes* throughout the procession of “salchichón” and the downregulation of the virulence gene expression of the surviving strains of this pathogen are relevant to control *L. monocytogenes* in this product. 

## Figures and Tables

**Figure 1 biology-10-01258-f001:**
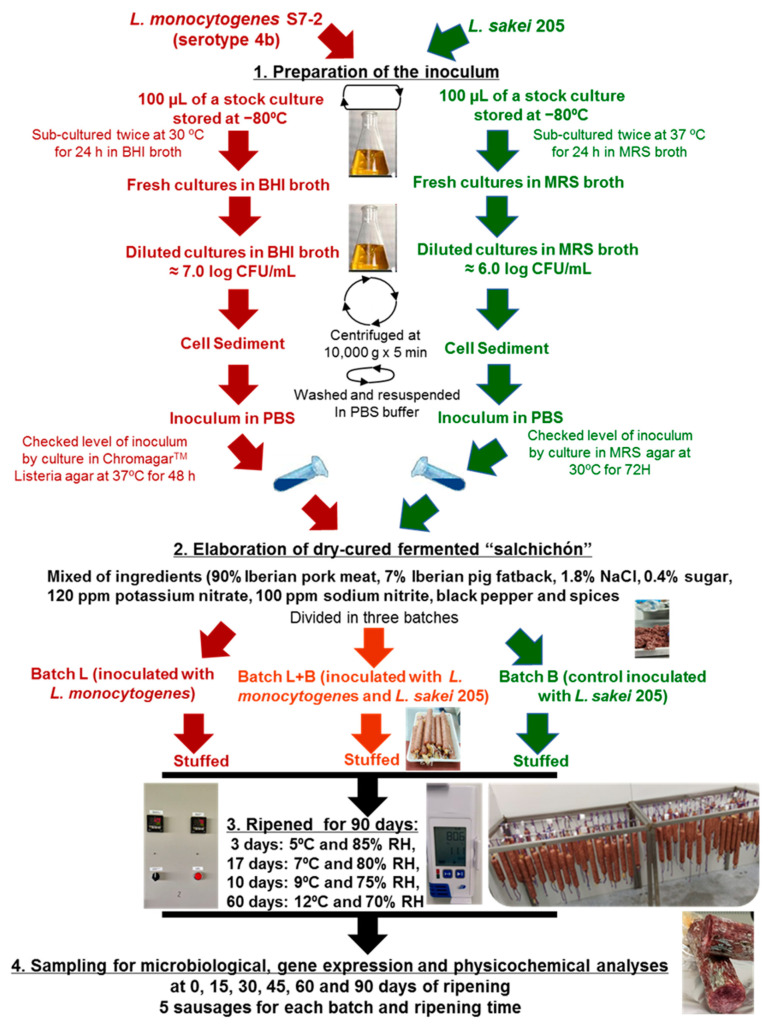
Scheme of the methodology followed in this work for preparing the inoculum, elaboration, and ripening conditions of the inoculated “salchichón” and sampling.

**Figure 2 biology-10-01258-f002:**
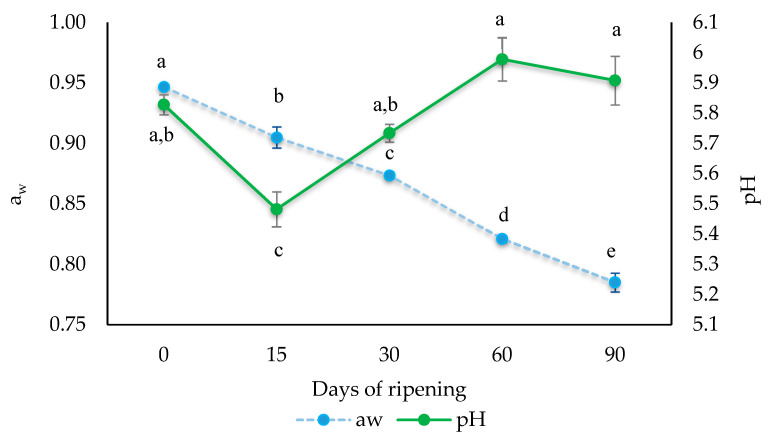
Evolution of water activity (a_w_) and pH of dry-cured fermented sausages “salchichón” throughout the ripening process. Different letters indicate significant differences in the same parameter at the different ripening days (*p* ≤0.05).

**Figure 3 biology-10-01258-f003:**
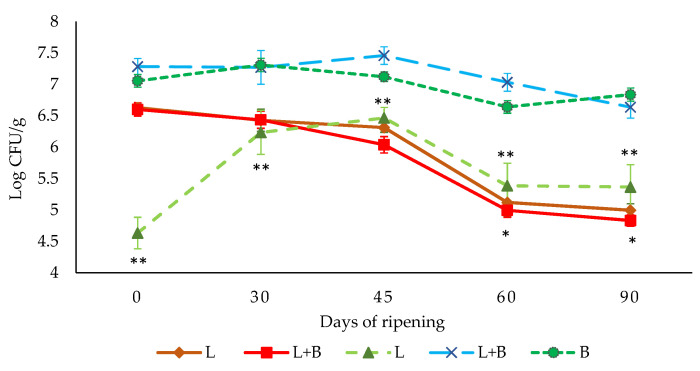
Growth curves of the LAB populations (dashed line) during the ripening process of dry-fermented sausages for batch B (●), L (▲) and batch L + B (X) and *L. monocytogenes* populations (solid line) for batch L (♦) and batch L + B (▪). (*) indicates significant differences (*p* ≤ 0.05) between batches of *L. monocytogenes* and (**) between batches of LAB.

**Table 1 biology-10-01258-t001:** Absolute expression values (mean log copies of gene ± standard deviation) of the virulence (*plcA*, *hly*, and *iap*) genes of *Listeria monocytogenes* in dry-cured fermented sausages “salchichón” throughout the ripening process. Batches: L: *L. monocytogenes*, L + B: *L. monocytogenes* + *Lactilactobacillus sakei* 205. The assays were conducted in quintuplicate.

Genes	Batches	Days of Ripening				
		0	15	30	60	90
*hly*	L	3.01 ± 0.372 ^1^	3.07 ± 0.099 ^1^	3.26 ± 0.216 ^1^	1.89 ± 0.226 ^2,b^	1.82 ± 0.370 ^2^
	L + B	3.28 ± 0.132 ^1,2^	2.96 ± 0.180 ^2^	3.42 ± 0.208 ^1^	2.48 ± 0.272 ^3,a^	1.49 ± 0.316 ^4^
*plcA*	L	4.17 ± 0.286 ^1^	4.32 ± 0.282 ^1,a^	4.41 ± 0.410 ^1^	2.47 ± 0.236 ^2^	3.06 ± 0.505 ^2^
	L + B	4.01 ± 0.265 ^1^	3.74 ± 0.440 ^1,b^	4.35 ± 0.271 ^1^	2.91 ± 0.494 ^2^	2.68 ± 0.483 ^2^
*iap*	L	2.72 ± 0.181 ^1^	2.69 ± 0.067 ^1^	2.82 ± 0.283 ^1^	1.65 ± 0.368 ^2^	1.96 ± 0.249 ^2^
	L + B	2.92 ± 0.165 ^1^	2.70 ± 0.076 ^1^	3.16 ± 0.378 ^1^	1.66 ± 0.433 ^2^	1.53 ± 0.491 ^2^

Within column, different superscript letters denote significant differences for the same gene in each batch at the different ripening days (*p* ≤ 0.05). Within row, different superscript numbers denote significant differences for the same gene and batch at each ripening day (*p* ≤ 0.05).

## Data Availability

Not applicable.

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
