# Peer review of "Growth and Expression of Virulence Genes of Listeria monocytogenes during the Processing of Dry-Cured Fermented “Salchichón” Manufactured with a Selected Lactilactobacillus sakei"

_biology, 2021, doi:10.3390/biology10121258_

Round 1

Reviewer 1 Report

The study entitled "Growth and expression of virulence genes of Listeria monocyto-2 genes during the processing of dry-cured fermented salchichon manufactured with a selected Lactilactobacillus sakei" deals with the effect of the incorporation of Lactobacillus sakei on survival and virulence of Listeria monocytogenes during the manufacture of a fermented meat product (Salchichón). It is an interesting topic, mainly the aspect related to the virulence of listeria.

However, there are shortcomings in the experimental design that force me to recommend the non-publication of this study.

Here are the following comments:

- Line 125. Because the sausage is produced by the presence of a natural microbiological flora present in the raw material and the environment, the study of sausage without listeria and without lactobacillus (i.e. batch C) should have been considered. In addition, the microorganisms present during the drying of the product should have been characterized.

- Line 141. The main deficiency of the study is the lack of repetitions. If no repetitions are performed, statistical analysis cannot be performed. At least three repetitions of each batch should have been done.

- Line 144. Where do you get the 10 g of salchichon? From the center? From the extremes? It should be better explained.

- Line 185. I do not understand the reason why the physical-chemical analysis is only carried out for the samples of batch B. It should have been carried out on all batches.

- Line 205. It is not possible to speak of significant differences without repetitions.

- Line 236. This statement (“From day 45 until the end of ripening, levels of L. monocytogenes were significantly lower in the batch inoculated with L. sakei 205 (L + B) than in 235 the batch only inoculated with L. monocytogenes”) cannot be sustained without an adequate statistical study.

Author Response

Reviewer #1:

The study entitled "Growth and expression of virulence genes of Listeria monocyto- genes during the processing of dry-cured fermented salchichon manufactured with a selected Lactilactobacillus sakei" deals with the effect of the incorporation of Lactobacillus sakei on survival and virulence of Listeria monocytogenes during the manufacture of a fermented meat product (Salchichón). It is an interesting topic, mainly the aspect related to the virulence of listeria.

Response:

We thank Reviewer 1 for these positive comments.

However, there are shortcomings in the experimental design that force me to recommend the non-publication of this study.

Here are the following comments:

- Line 125. Because the sausage is produced by the presence of a natural microbiological flora present in the raw material and the environment, the study of sausage without listeria and without lactobacillus (i.e. batch C) should have been considered. In addition, the microorganisms present during the drying of the product should have been characterized.

Response: The Lactilactobacillus sakei used in the present work was selected by its anti-L. monocytogenes activity in culture medium elaborated with “salchichón” in sterilized conditions. In the present work, the purpose was mainly to evaluate the effect of the selected L. sakei on the expression of virulence genes of L. monocytogenes in a real food system (dry-cured fermented “salchichón”) in presence of the natural microbiological population of this kind of product. For this, only batch B (control inoculated only with L. sakei), batch L (inoculated only with L. monocytogenes) and batch L + B (inoculated with L. monocytogenes and L. sakei) are necessary. According to Reviewer 1’s comment about the characterization of the microorganisms present during drying, this was carried out. In this study, 50% of the characteristic lactic-acid bacteria colonies were randomly isolated from MRS plates and characterized in all batches at the last sampling time (90 days) by sequencing analysis of the 16S rRNA region and PFGE analysis of the DNA with the restriction NotI and SgsI enzymes (Thermo Fisher Scientific, USA) following the procedures previously described by Martín et al. (2021). In batches inoculated with L. sakei (B and L+B) most of the tested isolates (86%) were identified as L. sakei (100% identity) by sequencing analysis of the 16S rRNA region and in the PFGE analysis these isolates showed the same pattern of L. sakei 205. The remaining strains were Lactilactobacillus plantarum group (7%) and Lactilactobacillus curvatus (7%). In batch L, none of the isolates was identified as L. sakei. Thus, the inoculated L. sakei 205 was the predominant strain in L. sakei-inoculated bathes (B and B+L) and it was not detected in batch inoculated only L. monocytogenes (L). This has been included in Materials and Methods section (page 3, lines 101-104; page 5, lines 165-169) and Results and Discussion section (page 7, lines 239-245) of the revised version of the manuscript.

Line 141. The main deficiency of the study is the lack of repetitions. If no repetitions are performed, statistical analysis cannot be performed. At least three repetitions of each batch should have been done.

Response:

The experiment consisted of 3 different batches x 5 sampling times x 5 different analyzed sausages/each batch and sampling time. This was done in accordance to the European Union Reference Laboratory Technical Guidance Document for conducting shelf-life studies on L. monocytogenes in RTE foods (such as “salchichón”) where no growth or the growth probability of this pathogen is ≤ 10% (Beaufort et al., 2014). This paragraph has been included in the revised version of the manuscript (page 5, lines 147-152). Furthermore, the processing of dry-cured fermented sausage “salchichón” was performed according to the industrial procedure for this product, and the evolution of the physicochemical parameters was consistent with those observed in other dry-cured fermented sausages (Fernández-López et al. 2008. Meat Sci. 2008, 80, 410–417; Metaxopoulos, et al. 2001. Ital. J. Food Sci. 2001, 13, 3–18). Thus, it is not expected in RTE products such as “salchichón” processed under industrial conditions, that successive repetitions of the entire experiment (with the abovementioned batches, sampling times and sausages), could lead to results different from those obtained in the present work.

- Line 144. Where do you get the 10 g of salchichon? From the center? From the extremes? It should be better explained.

Response:

For microbial analysis at each sampling time, 10 g of each of the 5 dry-cured sausages “salchichón” were aseptically taken from the center of the product and mixed with 90 mL of 1 % (w/v) peptone water and homogenized in a Stomacher machine. This has been included in the revised version of the manuscript (page 5, line 156).

- Line 185. I do not understand the reason why the physical-chemical analysis is only carried out for the samples of batch B. It should have been carried out on all batches.

Response:

Batch B was the only one non-inoculated with L. monocytogenes. Our purpose was to know the effect of L. sakei 205 on the physical-chemical parameters in comparison with previously published data for this product. In addition, due to safety reasons, only non-inoculated samples with pathogens could be tested for physical-chemical analysis in the corresponding equipment in the laboratory. 

- Line 205. It is not possible to speak of significant differences without repetitions.

- Line 236. This statement (“From day 45 until the end of ripening, levels of L. monocytogenes were significantly lower in the batch inoculated with L. sakei 205 (L + B) than in 235 the batch only inoculated with L. monocytogenes”) cannot be sustained without an adequate statistical study.

Response:

As it has been previously stated, the experiment consisted of 3 different batches x 5 sampling times x 5 different analyzed sausages/each batch and sampling time and was done according to the European Union Reference Laboratory Technical Guidance Document for conducting shelf-life studies on L. monocytogenes in RTE foods (such as “salchichón”) where no growth or the growth probability of this pathogen is ≤ 10%. The statistical analysis for each analyzed parameter was carried out with the values obtained for the 5 different analyses performed at each sampling time for each of the 3 different batches. Thus, from our point of view, the statistical analysis was appropriately performed and well-described in Material and Methods section. Besides, results and conclusion are well supported for this statistical study.

Reviewer 2 Report

Additional comments to authors:

 The study deals with the effect of the incorporation of Lactobacillus sakei on the survival and virulence of Listeria monocytogenes during the manufacture of a fermented meat product (Salchichón). The article also evaluates, the virulence aspect of listeria.

 It is an interesting paper mainly since sausage is produced by natural microbiological bulk during its industrial processing. The results were addressed by analyzing physicochemical, microbial and gene expression in a timeline way given a view of the analytical determinations in time.

 The authors have explained all of their findings, and their methodology, results, and conclusions are based on the evidence collected.  Although the topic is not new  I consider their proposal original.

 The main contribution is the fact that the study was conducted using sausage produced in a pilot industrial plant, which is very interesting since the results suggest control of the prevalence of L. monocytogenes in the process.

 The text has some English errors that must be corrected.

 The article is well structured but I would like to suggest a figure illustrating the methodology as it has several interconnected steps this will allow an overview of the methodology.

 The conclusions respond to the objectives but the authors did not explore all the results found so I suggest that the conclusion be increased.

 The scientific question is clear in the text.

=========================

The authors should place error bars in the graph in Figure 2 for an overview of the experimental errors involved.

Author Response

Reviewer #2:

To authors:

The study deals with the effect of the incorporation of Lactobacillus sakei on the survival and virulence of Listeria monocytogenes during the manufacture of a fermented meat product (Salchichón). The article also evaluates, the virulence aspect of listeria.

It is an interesting paper mainly since sausage is produced by natural microbiological bulk during its industrial processing. The results were addressed by analyzing physicochemical, microbial and gene expression in a timeline way given a view of the analytical determinations in time.

The authors have explained all of their findings, and their methodology, results, and conclusions are based on the evidence collected. Although the topic is not new I consider their proposal original.

The main contribution is the fact that the study was conducted using sausage produced in a pilot industrial plant, which is very interesting since the results suggest control of the prevalence of L. monocytogenes in the process.

Response:

We thank Reviewer 2 for the positive comments of the manuscript.

The text has some English errors that must be corrected.

Response:

According to Reviewer 2’s comment, English language has been revised in the new version of the manuscript.

The article is well structured, but I would like to suggest a figure illustrating the methodology as it has several interconnected steps this will allow an overview of the methodology.

Response:

Following the recommendation of Reviewer 2, a new Figure (Figure 1) illustrating the methodology has been included for a better understanding for the readers. Thus, Figures 1 and 2 of the previous version of the manuscript are now Figures 2 and 3 in the revised version.

The conclusions respond to the objectives, but the authors did not explore all the results found so I suggest that the conclusion be increased. The scientific question is clear in the text.

Response:

The conclusions have been rewritten and increased with the purpose to explore all the results found in this paper.

The authors should place error bars in the graph in Figure 2 for an overview of the experimental errors involved.

Response:

We thank Reviewer 2 for this suggestion. The error bars have been included in Figure 3 and also in Figure 2 (Figures 1 and 2 of the previous manuscript), of the revised version of this paper.

Reviewer 3 Report

Manuscript ID: biology-1429223-peer-review-v1

Growth and expression of virulence genes of Listeria monocytogenes during the processing of dry-cured fermented “salchichón” manufactured with a selected Lactilactobacillus sakei

I think it's a very interesting and very important topic for food hygiene and technology nowadays. The contamination of L. monocytogenes of RTE foods and challenge test are object of studies in the food safety field. The topic is of interest for the academics and for the industry because of the results obtained and because of the novelty of the application in field. There are some studies like this in literature, but not specific in this kind of RTE product.

The manuscript evaluate the presence and contamination of “salchichon” by L. monocytogenes during ripening after inoculation of starter LAB; the research is well performed, the sampling and analysis were well done.

The manuscript is well written and easy to understand by readers. I believe that this manuscript does not need big changes and I think you can publish the manuscript in its current form after few minor revision.

Statistical analysis was well performed

Results and Discussion were well explained and the conclusion are of interest.

Author Response

Reviewer #3:

I think it's a very interesting and very important topic for food hygiene and technology nowadays. The contamination of L. monocytogenes of RTE foods and challenge test are object of studies in the food safety field. The topic is of interest for the academics and for the industry because of the results obtained and because of the novelty of the application in field. There are some studies like this in literature, but not specific in this kind of RTE product.

The manuscript evaluate the presence and contamination of “salchichon” by L. monocytogenes during ripening after inoculation of starter LAB; the research is well performed, the sampling and analysis were well done.

The manuscript is well written and easy to understand by readers. I believe that this manuscript does not need big changes and I think you can publish the manuscript in its current form after few minor revision.

Statistical analysis was well performed

Results and Discussion were well explained, and the conclusion are of interest.

Response:

We thank Reviewer 3 for all these positive comments.

Since Reviewers 2 and 3 proposed that English language and style are fine/minor spell check, English style has been corrected in the revised version of the manuscript.
